# MPA_Pathway_Tool: User-Friendly, Automatic Assignment of Microbial Community Data on Metabolic Pathways

**DOI:** 10.3390/ijms222010992

**Published:** 2021-10-12

**Authors:** Daniel Walke, Kay Schallert, Prasanna Ramesh, Dirk Benndorf, Emanuel Lange, Udo Reichl, Robert Heyer

**Affiliations:** 1Bioprocess Engineering, Otto von Guericke University, Universitätsplatz 2, 39106 Magdeburg, Germany; kay.schallert@ovgu.de (K.S.); benndorf@mpi-magdeburg.mpg.de (D.B.); emanuel.lange@ovgu.de (E.L.); ureichl@mpi-magdeburg.mpg.de (U.R.); 2Database and Software Engineering Group, Otto von Guericke University, Universitätsplatz 2, 39106 Magdeburg, Germany; prasanna.ramesh@ovgu.de; 3Applied Biosciences and Process Engineering, Anhalt University of Applied Sciences, Microbiology, Bernburger Straße 55, 06354 Köthen, Germany; 4Bioprocess Engineering, Max Planck Institute for Dynamics of Complex Technical Systems, Sandtorstraße 1, 39106 Magdeburg, Germany

**Keywords:** omics, web application, pathway generation, pathway mapping, metaproteomics, bioinformatics

## Abstract

Taxonomic and functional characterization of microbial communities from diverse environments such as the human gut or biogas plants by multi-omics methods plays an ever more important role. Researchers assign all identified genes, transcripts, or proteins to biological pathways to better understand the function of single species and microbial communities. However, due to the versality of microbial metabolism and a still-increasing number of newly biological pathways, linkage to standard pathway maps such as the KEGG central carbon metabolism is often problematic. We successfully implemented and validated a new user-friendly, stand-alone web application, the MPA_Pathway_Tool. It consists of two parts, called ‘Pathway-Creator’ and ‘Pathway-Calculator’. The ‘Pathway-Creator’ enables an easy set-up of user-defined pathways with specific taxonomic constraints. The ‘Pathway-Calculator’ automatically maps microbial community data from multiple measurements on selected pathways and visualizes the results. The MPA_Pathway_Tool is implemented in Java and ReactJS.

## 1. Introduction

In the last years, studying the taxonomic and functional composition of bacterial, viral, and archaeal species in the human gut [1,2,3,4,5], in biotechnology [6,7,8], or the environment [9,10] has improved our understanding about diseases and the actual microbial process within these ecosystems [8]. There are several different approaches for analyzing microbial communities, focusing on the entirety of the genes (metagenomics), transcripts (metatranscriptomics), or proteins (metaproteomics). Whereas metagenomics reveals only the presence of genes, and metatranscriptomics only the gene expression, metaproteomics indicates actual protein expression [8]. On the basis of the protein expression levels, a microbial communities’ phenotype can be linked with specific environmental conditions, process parameters, or diseases [11]. Due to the complexity and amount of multi-omics data, comprehensive bioinformatic workflows were developed for the data evaluation [12,13,14,15,16]. For example, the MetaProteomeAnalyzer (MPA) enables the analysis and and interpretation of metaproteomic data sets. It offers a free, open-source, end user-oriented complete pipeline from peak lists to taxonomic and functional result evaluation. Among others, the MPA links identified proteins to functional categories (e.g., biological keywords) and the KEGG pathways [13]. In addition to the KEGG pathway system [17], several other pathway collection and mapping tools such as Reactome [18], Escher [19], and Pathway Tools [20] exist, supporting the data analysis of omics-datasets. For more details about the assignment of genes and proteins to functions and pathways, please refer to Mao et al. (2005) [21].

However, due to the microbial metabolism’s versatility and constantly newly discovered biological pathways [22], linkage to standard pathway maps is insufficient for many microbial community studies. Therefore, new tools are required that are tailored for microbial community studies.

In general, a good pathway tool needs to meet at least the following six criteria: (i) It should provide an easy and intuitive creation of pathways to enable the fast generation of multiple pathways. (ii) Since new reactions are discovered and pathways might be updated, the tool should support modifying the pathway maps, i.e., appending new and deleting existent reactions. (iii) Already created pathways from different databases should be reusable. Consequently, the pathway tool should provide an import function for standard exchange formats, such as comma-separated values (CSV), JavaScript Object Notations (JSON), and Systems Biology Markup Language (SBML) formats. (iv) The pathway tool should also map experimental data on created pathways and highlight differences between the considered samples. (v) Since metabolic reactions are taxonomy-specific, the pathway tool needs a filter to distinguish between reactions carried out by a specific taxon. One example of this specificity is the hydrogenotrophic methanogenesis and the Wood–Ljungdahl pathway. Both pathways share similar enzymes (i.e., similar Enzyme Commission numbers (EC numbers)). However, hydrogenotrophic methanogenesis is carried out only by archaea [23], while the Wood–Ljungdahl pathway is carried out mainly by bacteria [8]. (vi) The tool should be independent of operating systems so that nearly everyone can use the tool, favoring an implementation as a web application.

This paper aims to present a new web application called MPA_Pathway_Tool. It enables easy creation of user-defined pathways with specific taxonomic constraints, automatically mapping microbial community data from multiple measurements on selected pathways and visualizing the results (‘Pathway-Creator’). Additionally, the ‘Pathway-Calculator’ enables the mapping of an entire omics data set on multiple pathways to support the automated data analyses. The functionality of the MPA_Pathway_Tool is demonstrated and validated by reproducing the manual assignment of proteins to metabolic pathways from a previous metaproteomics study about biogas plants, focusing on three pathways (hydrogenotrophic methanogenesis, acetoclastic methanogenesis, and Wood–Ljungdahl pathway).

## 2. Results and Discussion

### 2.1. The ‘Pathway-Creator’ Enables Users to Define Their Own Metabolic Pathways

The first part of the MPA_Pathway_Tool, the ‘Pathway-Creator’ (Figure 1), enables the creation of user-defined pathways by adding reactions iteratively and linking omics data to this pathway. The menu (left side) allows for uploading of experimental data and pathways (such as CSV, JSON, and SBML), adding new user-defined reactions or reactions from KEGG, and downloading created pathways (such as CSV, SBML, JSON, and Scalable Vector Graphics (SVG)) and mapped data (such as CSV). The right side visualizes the created pathway. Circular-shaped nodes (metabolites; in KEGG referred to as compounds) and diamond-shaped nodes (reactions) are connected by arrows displaying the direction of each reaction. After a sample of previously uploaded data is selected by clicking on the respective button (bottom side of the tool), reaction nodes are colored dependent on their abundance in the sample. Information about abundances in all samples for a specific reaction is available as a heatmap by clicking on the respective reaction node.

### 2.2. The ‘Pathway-Calculator’ Enables Automated Mapping of Experimental Data on Multiple Metabolic Pathways

The ‘Pathway-Calculator’ (Figure 2) consists of two upload zones, one for experimental data and another for multiple pathway files (such as CSV, JSON, or SBML). It performs mapping (details in Section 3.3) of experimental data on uploaded pathways. After experimental data are mapped on uploaded pathways, the resulting table can be downloaded as CSV. Furthermore, a list with all unmatched features (e.g., proteins) can be exported. Details about the taxonomic structure of data mapped on analyzed pathways can be found under ‘See Details’. Finally, we also provide metadata about the mapping containing among others mapping time, used experimental data, and pathways. It fulfills the FAIR (findable, accessible, interoperable, and reusable) principles [24] and highlights the importance of emerging metadata standards [25].

Finally, we tested the performance of the ‘Pathway-Calculator’ by uploading experimental data with different file sizes (10,000 proteins, 100,000 proteins, and 1,000,000 proteins with 44 samples per test) and a different number of pathways on a local desktop computer (AMD Ryzen 5 3600, 16 GB DDR4 RAM 3000 MHz, Chrome Browser version 89.0.4389.90). We utilized 1, 10, and 100 copies of the user-defined Wood–Ljungdahl pathway used in Heyer et al. [8] (Biogas plant (BGP)-Wood–Ljungdahl pathway) for the performance test. Each test with 10,000 and 100,000 proteins finished within 1 min, indicating a good performance for most files. Tests with 1,000,000 proteins took longer (up to 12 min) caused by higher upload times and high requirements on memory and on CPU performance (Table 1).

### 2.3. Tool Validation with Experimental Data

Validation of the MPA_Pathway_Tool was carried out by comparing our pathway assignment against a manual pathway assignment of a previous metaproteomics study regarding biogas plants [8]. Additionally, we assigned the proteins to the standard KEGG pathways (in KEGG referred to as KEGG MODULEs) to illuminate the demand for user-specific pathways.

For the evaluation, we created six pathway maps (BGP-hydrogenotrophic methanogenesis, BGP-acetoclastic methanogenesis, BGP-Wood–Ljungdahl pathway, KEGG-hydrogenotrophic methanogenesis, KEGG-acetoclastic methanogenesis, and KEGG-Wood–Ljungdahl pathway). The user-defined pathways (BGP; Appendix A) and the KEGG pathways (Appendix A) were created with the ‘Pathway Creator’ (further details in Appendix A). For each pathway, a taxonomic classification was added (Table 2). According to the publication of the dataset, we decided to exclude Archaea from the Wood–Ljungdahl pathway and include only Archaea for the hydrogenotrophic and acetoclastic methanogenesis [8]. Subsequently, the experimental metaproteomics data (Appendix A) were mapped on each pathway using multiple pathways (Figure 3; Appendix A) and single pathway mapping (Figure 4; Appendix A). Unmatched proteins were downloaded as CSV (Appendix A).

The summed spectral counts mapped on each user-defined pathway were slightly greater than in the published experimental data, indicating a small information loss in the previous publication (Appendix A).

Furthermore, we compared the results of each BGP pathway with the corresponding KEGG pathway. We observed higher spectral counts for the created KEGG MODULEs compared to the respective BGP pathways (Figure 3). Higher spectral counts in KEGG MODULEs are caused by an integration of more reactions in the KEGG MODULEs than in the BGP pathways (Figure 4). For example, KEGG-acetoclastic methanogenesis and KEGG-hydrogenotrophic methanogenesis possess three additional reactions (2.1.1.86, 2.8.4.1., and 1.8.98.1) that catalyze the last steps of the methanogenesis. BGP pathways include only pathway-specific reactions to prevent wrong pathway identification.

### 2.4. Conclusion: The MPA_Pathway_Tool Provides An Easy and Fast Option to Set up Multiple Pathways

We successfully implemented the pathway tool to meet all of our six defined criteria: (i) The MPA_Pathway_Tool provides an easy and fast setup of multiple pathways. Multiple reactions can be imported using various options, e.g., import by EC numbers, import of a KEGG MODULE, or import of entire SBML files. (ii) A further modification of the generated pathways is possible by deleting reactions and adding new reactions (from the KEGG database and user-defined reactions). (iii) As interchange formats, JSON, CSV, and SBML were implemented. (iv) Experimental data from metaproteomics, metatranscriptomics, and metagenomics studies can be automatically mapped on single pathways (‘Pathway-Creator’) and multiple pathways (‘Pathway-Calculator’). The results of pathway mapping can also be exported as CSV. (v) The mapping algorithm includes a taxonomic filter that was successfully applied by comparing our results with published experimental data for the stated pathways. (vi) The MPA_Pathway_Tool was implemented as a stand-alone web application to guarantee the independence from users’ operating systems.

### 2.5. Future Work: Integration to Other Features and Addition of New Functions Will Increase the Flexibility of the MPA_Pathway_Tool

The MPA_Pathway_Tool is developed as a stand-alone application for the characterization and analysis of microbial community data. To broaden its scope of application, it will be integrated into the next version of the MPA software. However, in the future, other tools such as UniPept [26] or Prophane [27] might also be linked.

Since the MPA_Pathway_Tool already provides stoichiometric data, further options such as identification-driven flux balance analysis and flux vector analysis might be added to the workflow using tools such as the CellNetAnalyzer [28], COBRApy [29], or Escher [19]. As it allows the calculation of metabolic flows in metabolic networks, it can be used to predict the specific growth rate of organisms and the specific production rate of certain metabolites. Accordingly, identification of strategies for optimizing biotechnological processes might be feasible [30].

## 3. Availability and Implementation

### 3.1. General Workflow

The MPA_Pathway_Tool represents an intuitive web application for mapping (meta)-proteome and other omics data on metabolic pathways and the creation and modification of pathways. It consists of two different parts. First, the ‘Pathway-Creator’ allows for the creation and modification of pathways. The ‘Pathway-Creator’ also supports single-pathway mapping. The second part is the ‘Pathway-Calculator’, which provides multiple pathway mapping.

### 3.2. Implementation

The MPA_Pathway_Tool is a stand-alone web application consisting of JavaScript (JS), Hypertext Markup Language 5 (HTML), and Cascading Style Sheets (CSS) on the client side and Java on the server side. The client side is built with the help of ReactJS [31]. We used Create React App for the setup of the project. Create React App is used to create single-page React applications, and it provides a setup without configurations [32]. Third-party dependencies have been pulled by using Node Package Manager (npm) (version 6.14.9). The most important dependencies are ReactJS (version 16.8.6) [31] for using JSX syntax in our project, Redux (version 4.0.5) [33] and mobx (version 7.0.5) [34]—used for storing and handling states, React-d3-graph (version 2.5.0)—used for visualization of pathways [35], Axios (version 0.21.0) [36]—used for making HTTP requests from the browser, Material-UI (version 4.11.2) [37]—used for the implementation of some user-interface components, Lodash (version 4.17.21) [38]—used for deep cloning objects, and File-saver (version 2.0.5) [39]—used for saving files.

A complete list of used dependencies on the client side is listed in the package.json file (https://github.com/danielwalke/MPA_Pathway_Tool/tree/main/keggcalculator-frontend, accessed on 8 October 2021).

We implemented a REST-API (Figure 5) and an algorithm for mapping data on multiple pathways in the programming language Java (Java SE-1.8). Dependencies were imported using Maven. We used Gson (version 2.8.5) [40] for converting Java objects to JSON and vice versa, Spark (version 2.6.0) for setting up the REST-API, and JSBML (version 1.5) [41] for reading SBML files.

Implemented exchange formats for created metabolic pathways include CSV, JSON, and SBML. In addition to the metabolic reactions, we stored in these files the opacity of nodes, their position, and the abbreviations for later usage or modifications of the pathways. This information is stored in a specific tag (‘layout:listOfLayouts’) of SBML files (Figure 6).

### 3.3. Mapping Algorithm

For a better understanding of the mapping algorithm, we explain the application with experimental data containing proteins. Of course, the mapping algorithm can be applied for other -omics data too. Each pathway contains a pathway name (filename) and multiple reactions. On the other hand, each reaction contains an ID; a name; substrates, and products with stoichiometric coefficients, EC-numbers, K-numbers, and taxonomic constraints; and a list of matching proteins, which is empty at the beginning of the mapping calculation. For a better overview, only the last four properties are visualized as a scheme (Figure 7). The imported file with protein identifications contains multiple proteins, which comprise an identifier, EC-numbers, K-numbers, a taxonomic tree (superkingdom, kingdom, phylum, class, order, family, genus, species), a description, and quantified values for each measured sample (Appendix A, example input). For each reaction in a pathway, all proteins are analyzed. For a protein to be added to the list of matched proteins, two conditions must be fulfilled. The first condition is fulfilled if the EC-numbers or K-numbers of the reaction contain at least one EC-number or K-number of the protein. The second condition is fulfilled if the taxonomic tree of the protein contains the taxonomic requirement specified for the reaction. The fulfilment of both conditions adds the protein to the list of matched proteins. The quantitative values of all proteins are summed up for each sample. For single pathway mapping, the results can be visualized on the created pathway and exported as CSV. For multiple pathway mapping, all quantified values of each reaction (sums of quantified values of matched proteins) are summed up. Additionally, each pathway’s completeness is checked by returning the number of reactions identified in the data and the total number of reactions in the pathway. Finally, results can be exported as CSV. Multiple pathway mapping also provides the calculation of multiple imported pathways. 

### 3.4. Experimental Data

Experimental data used for validating the MPA_Pathway_Tool originated from a previous metaproteomics study of 11 different biogas plants sampled at two different time points [8]. As input, we used the final result matrix of all identified metaproteins (see Appendix A, in [8]).

### 3.5. Availability

The MPA_Pathway_Tool is freely available on the web at https://mpa-pathwaymapper.ovgu.de/ or http://141.44.141.132:9001/home. Further documentation and the complete source code are deposited on GitHub (https://github.com/danielwalke/MPA_Pathway_Tool, accessed on 8 October 2021).

## Figures and Tables

**Figure 1 ijms-22-10992-f001:**
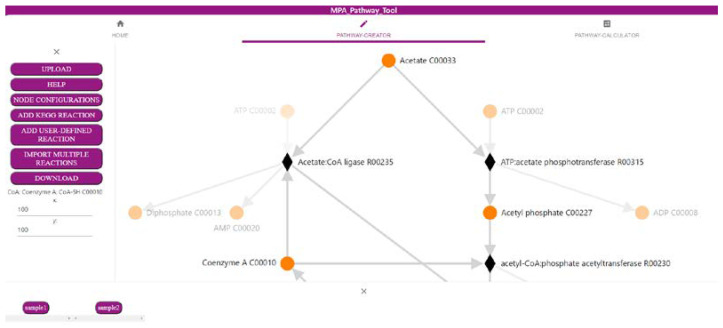
Screenshot of the ‘Pathway-Creator’ of the MPA_Pathway_Tool (details in Section 2.1).

**Figure 2 ijms-22-10992-f002:**
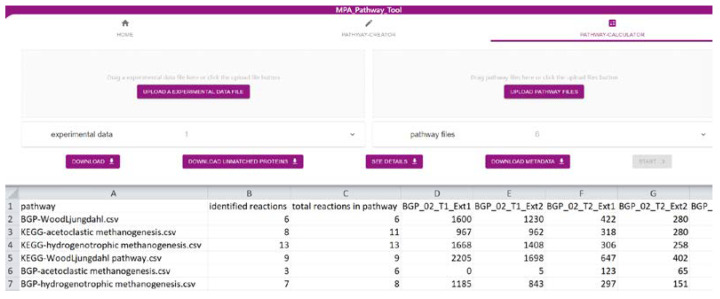
Screenshot of the ‘Pathway-Calculator’ of the MPA_Pathway_Tool (details in Section 2.2).

**Figure 3 ijms-22-10992-f003:**
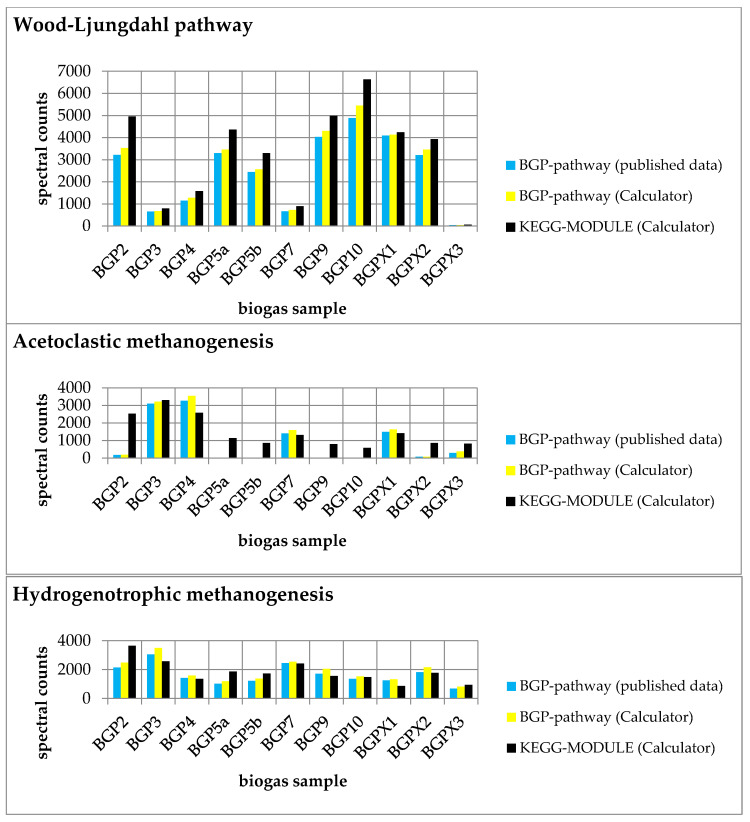
Comparison of experimental data mapped on each pathway (Wood–Ljungdahl pathway, acetoclastic methanogenesis, and hydrogenotrophic methanogenesis) with published data [8]. Summed spectral counts plotted against all biogas samples are blue and yellow for user-defined pathways and black for KEGG MODULEs. The results were obtained from the ‘Pathway-Calculator’ (yellow and black) and published data (blue). In particular, acetoclastic methanogenesis showed higher spectral counts in most samples, indicating the occurrence of unspecific reactions.

**Figure 4 ijms-22-10992-f004:**
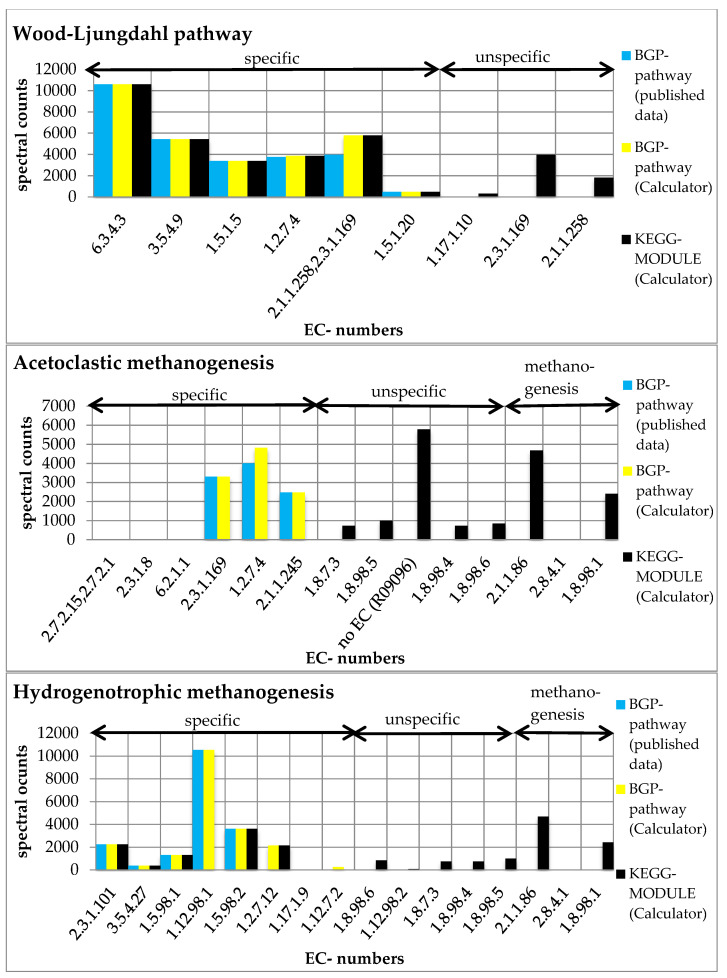
Summed spectral counts of each reaction in the Wood–Ljungdahl pathway, acetoclastic methanogenesis, and hydrogenotrophic methanogenesis. Results from published data [8] are visualized in blue, results from the ‘Pathway-Calculator’ that are user-defined are shown in yellow, and results from the ‘Pathway-Calculator’ for KEGG MODULEs are shown in black. All KEGG MODULEs possess additional unspecific reactions. KEGG-acetoclastic methanogenesis and KEGG-hydrogenotrophic methanogenesis possess three additional reactions (2.1.1.86, 2.8.4.1., and 1.8.98.1) that catalyze the last steps of the methanogenesis (see Appendix A).

**Figure 5 ijms-22-10992-f005:**
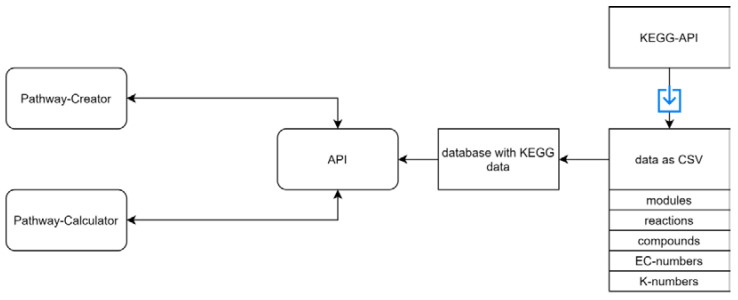
An overview of the REST-API. A REST-API is used for sending the required data to the ‘Pathway-Creator’ and the ‘Pathway-Calculator’. The KEGG database was downloaded as CSV files and parsed into our database. The available KEGG data include pathway modules, compounds, reactions, EC-numbers, and K-numbers. The blue arrow represents a download of specific files.

**Figure 6 ijms-22-10992-f006:**
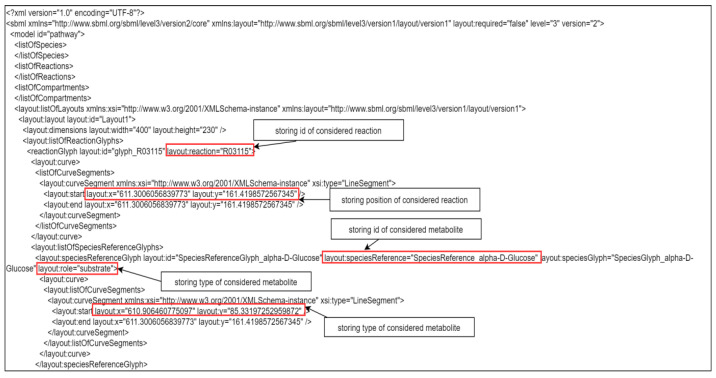
An example of updates in an SBML file. In the tag ‘layout:listOfLayouts’, all positions for the nodes of the created pathway are stored.

**Figure 7 ijms-22-10992-f007:**
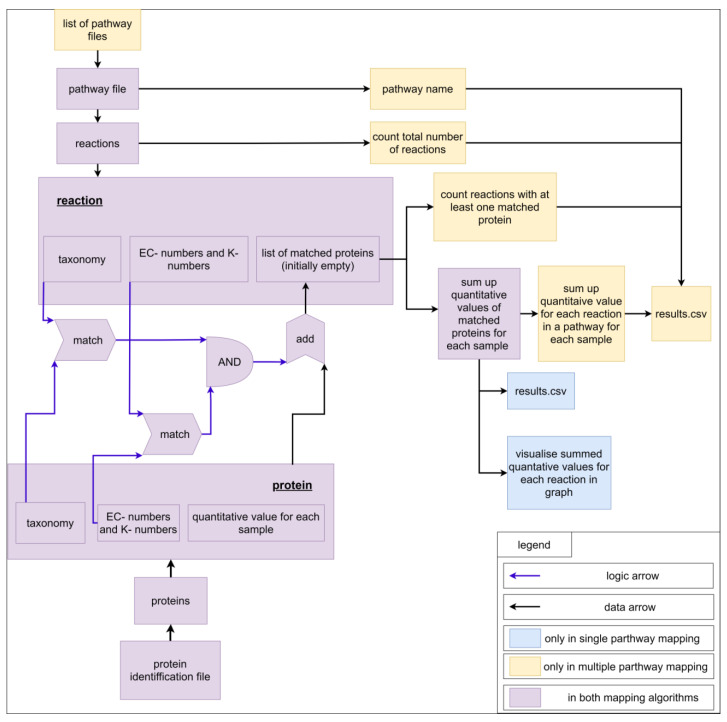
Scheme for visualization of the mapping algorithm.

**Table 1 ijms-22-10992-t001:** Performance test for the ‘Pathway-Calculator’ (details in Section 2.2).

Experimental Data	1 Pathway	10 Pathways	100 Pathways
10,000 proteins	5 s	5 s	6 s
100,000 proteins	5 s	10 s	36 s
1,000,000 proteins	192 s	244 s	712 s

**Table 2 ijms-22-10992-t002:** Taxonomic classification of each pathway.

Pathway	Added Taxonomic Requirement
BGP-hydrogenotrophic methanogenesis (user-defined)	only Archaea
KEGG-hydrogenotrophic methanogenesis (KEGG-MODULE)	only Archaea
BGP-acetoclastic methanogenesis (user-defined)	only Archaea
KEGG-acetoclastic methanogenesis (KEGG-MODULE)	only Archaea
BGP-Wood–Ljungdahl pathway (user-defined)	all except Archaea
KEGG-Wood–Ljungdahl pathway (KEGG-MODULE)	all except Archaea

## Data Availability

We included all data on supplements and on GitHub (https://github.com/danielwalke/MPA_Pathway_Tool, accessed on 8 October 2021).

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
