# Peer review of "MPA_Pathway_Tool: User-Friendly, Automatic Assignment of Microbial Community Data on Metabolic Pathways"

_ijms, 2021, doi:10.3390/ijms222010992_

Round 1
Reviewer 1 Report
Walke et al. describe a web-based software application, the MPA_Pathway_Tool, to infer potential metabolic pathways among microbial community data (such as metagenomics & metaproteomics studies of microbiomes).
As the authors have noted, this is an important research area which is often challenging for a variety of reasons. The first factor-- which could perhaps be emphasized in the text a bit more-- is ease of use. The MPA_Pathway_Tool appears to provide a simple method for end-users to interact with complex data tiers and multiple measurements.
In addition, knowledge of microbial metabolism is a dynamically changing landscape and will continue to change with the suite of analytical technologies available to the scientific community. While I was not able to directly access the web-page where the tool is hosted, I was able to see that the application's source code has been posted to Github.
REQUESTED MODIFICATIONS & CHANGES: Provided the four (04) items are addressed below, I recommend this paper for acceptance after reviewing the authors' responses to my review.
01. From Lines 49-51, the authors highlight many software tools for data analysis, including Reference 12, which describes the Galaxy platform. As the authors demonstrate their tool's applicability on a metaproteomics dataset (of their own), I would ask that they include a citation of the GalaxyP (Galaxy Proteomics) platform of software tools:
Blank C, et al. Disseminating Metaproteomic Informatics Capabilities and Knowledge Using the Galaxy-P Framework. Proteomes. 6: 7. 2018
URL: https://www.mdpi.com/2227-7382/6/1/7
DOI: https://doi.org/10.3390/proteomes6010007
02. In manuscript lines 60-61, the authors bring to light additional criteria for "good pathway tools." The authors mention importing _experimental_ details of data acquisition by JSON, CSV, and SBML formats (point iii.), which is incredibly important. I applaud the authors for speaking to the importance of metadata among these analyses; however, there appear to be 2 disconnects that could be addressed:
A. Recording MPA Pathway Tool mapping steps in it own metadata, of as a so called "computational audit trail," if you will. Building in such a capability would, for example, provide a documented record of what steps were performed in the data analysis, what parameters were used from experimental data, which user(s) performed the data analysis, and what version (or versions) of pathway data were experimental data mapped to.
Concatenating a data analysis audit trail through the MPA Pathway Tool fulfills prerequistes of reproducible research, FAIR Principles for research (www.go-fair.org), and emerging metadata standard initiatives among metagenomics & metaproteomics communities.
B. Providing a direct link between experimental measurement metadata (which is referred to in point iv.); however, this could also tie in to exisiting experimental metadata standards among sequence repositories (GEO, EMBL, etc.) and proteomics data repositories (PRIDE, proteomeXchange).
Recent work towards establishing metadata standards at this level of detail been reported in both research communities. Including a citation of the following work regarding emerging metadata standards would benefit the audience of this manuscript. As such, please cite: Vangay P, et al. mSystems. 2021. DOI: https://doi.org/10.1128/mSystems.01194-20
From lines 209-214, the authors spend a considerable effort in explaining how and why previous results of this MPA Pathway mapping differ from their previously published results. This is understandable in some respects, as they authors have mentioned that putative pathway data is iteratively updated and some EC numbers have been depreciated. As I have mentioned above, if the authors were to automatically record a "computational audit trail" of experimental parameters, KEGG versions, and other metadata relating to the previously published results, it would not be necessary to devote 15 lines of text to justify observed differences. The authors could simply include a TXT or JSON file as Supplementary Information, for example.
03. For testing, evaluation, and to show proof-of-concept, the authors use one of their own previous datasets: Reference 8, Heyer R, et al. Microbiome. 2019. https://microbiomejournal.biomedcentral.com/articles/10.1186/s40168-019-0673-y . While understandable given the lab's expertise, it would have been worthwhile to see how the tool performs among other microbiome data (eg. only metagenomics) or data acquired outside of their own research group.
04. If I have any objections to what is described in the manuscript, they are only semantic in nature and frustratingly pervasive in omics research. As stated in my first paragraph (review) of the manuscript, the authors have developed a tool to infer potential metabolic pathways in microbial consortia. If these pathways are being mapped to proteins identified from a predicted proteome, then that protein's function is likely inferred from electronic annotation (eg. sequence similarity). I would venture to say that the majority of the output are in silico predictions of metabolic pathways rather than actual "pathway assignments." I think that it is ever-increasingly important to call things what are as accurately as possible in the context of use, that's all.
MINOR EDITS AND SPELLING CHECKS:
Potential typo or error in Table 1 heading, line 139: "...time in seconds beTable 5." Please correct this, as it looks as though "beTable 5" may be a typo for the CPU that is described above; this is unclear and confusing in review draft form, proof, and or type-setting.
The Figure 1 legend is far too long and includes many potential typographical errors (such as punctuation in the middle of putative functional/pathway assignments). Please condense these and/or display these results in another format, such as a table; and, correct the typographical errors in the legend.
Reviewer 2 Report
Within their study "MPA_Pathway_Tool: User-friendly, automatic assignment of microbial community data on metabolic pathways" Walke et al. present a stand alone tool for metabolic pathway administration and meta-proteome data2pathway mapping and summarizing. Maybe I overlooked something, the given link for accessing MPA_Pathway_tool (MPAPT) online did not work - so I will stick to the information of the presented manuscript. MPAPT consists of two parts - the pathway creator and the pathway calculator. While the first one is an administrative tool for pathway data administration the second one really does the work and sums up quantitative values from meta-proteome files according pathway assignments based on EC- and KO-numbers (is it really K-numbers?). Technically the authors considered established standards and APIs. KEGG modules can be imported, SBML ist supported, CSV files can be read and written, SVG and PNG for graphics files is supported. If this works online, the tool may be useful and would be an interesting add on to MPA or other meta-proteome analysis tools such as PROPHANE or others. But is this sufficient for an individual paper? Put maliciously: The paper could have been written by a perfectly capable master's student with special interest in (bio)informatics who was either left to his own devices or got bogged down or didn't listen to hints and then ran out of time. But my job is not to conjecture. So back to the facts. In general, the English style and grammar should be improved. Please condense the text and be less redundant and more straight forward. Especially 2.1 and 2.2 as well as 3.3 and 3.4 often present the same facts. some inaccuracies: line 44: increasingly important - for what and for whom? line 46-47: metatranscriptomics and metaproteomics indicate actual protein expression - this is not correct - metatranscriptomics gives an overview on active genes, or genes currently transcribed. For RNA-genes no proteins will be synthesized, for resulting mRNA with posttranscriptional silencing or degradation also no proteins will be synthesized. If a protein has been detected (eg by metaproteomcis), this is the result of protein synthesis (translation or also called gene expression - there is no protein expression) AND protein modification/post translational modification, protein targeting and transport processes, partly protein cleavage or complete degradation, precipitation, storage in inclusion bodies etc pp and also translation free protein synthesis - some peptide antibiotics. line 72: carried out by a specific taxonomy -> taxon line 130: and different (?) a number of pathways Fig 3 and 1 in wrong order line 213: corrected -> circumvented line 55-56: Reactome [18], Pathview Web [19] - is the same as or is based on KEGG, Escher [20], Pathway Tools [21] I liked the clear statement of six points, that should be fullfilled Think about the priorities, by my opinion the reuse and import is the most improtant point (i.) easy and intuitive creation of (multiple) pathways (ii.) modifying pathway maps (iii.) use pathways from different databases (import for CSV, JSON, SBML) (iv.) mapping and highlighting metaproteome data from different experiments (v.) taxon-specific filtering (vi.) cross platform implementation Results In the result section pathway creator and calculator are presented The pathway creator provides both pathway editing and data mapping. Due to accession problems I was not able to test, how usable pathway editing is. Considering rules in data visualization there is room for improvements: Put the logos on a footer, you do not need to scroll for starting your work The most valuable ressource in visualization is screen space, do not waste it. Make it possbile to load example data by clicking an example button or link Keep the layout as clear as possible slim arrowhead that do not interfere with geometrical objects I suggest to replace the diamond symbols with squares (the edges of the diamonds increase visual complexity due to their unparallelity) Please postion the labels in a way, that they do not interfere or overlap with lines Or use automatic layouts (see yfiles) or cytoscape Support the user with help links (small question marks popping up tool tips) How the color mapping is performed, which type of data is supported (programm or scientist generated ratios, log2ratios, z-scores?) Is it possible to set arrowheads on both sides for equilibrium reactions? Can you remove the arrowhead near the enzyme nodes? Instead (C means Compound, E means Enzyme) (C)->[E]->(C) only (C)-[E]->(C) (C)<-[E]<-(C) only (C)<-[E]-(C) (C)<-[E]->(C) for equilibrium reactions How the abundances are represented after clicking an enzyme node? is it bar charts, a heat map row of colors, or symbols of different sizes can you distinguish, which taxon contributes most rows of the 10 most abundant taxa + others in sum as microcharts How the uploaded data need to be structured for proper processing? Is the uploaded data the same as in the pathway calculator? The pathway calculator In the pathway calculator it seems, that the pathways are interpreted as groups of entities and the abundances of the identified proteins from the MPA file are summed up according their membership this groups (based on EC or KO number). As an output a table is generated, there Enzymes of a pathways are listed by count, mapped enzymes are given as count and the summed up abundances of all Enzymes of a pathway for any sample are given in sample columns. Wouldn't it be nice to have the possibility to see the taxon distribution for the summed up abundances per pathways by clicking on a row a get a table only representing one pathway and different taxa in rows with their taxon specific summed up abundances to see which taxonomic groups contributes most to a pathway within the whole metasystem. Performance test was done, this is good. Data were tested on a real lab sample collection. I think the taxonomic assignment of pathways is a crux and may work especially in this individual scenario. I like my suggested detailed representation on a taxon level a lot more. This would work without prior knowledge about which taxa perform which pathways. Such knowlege in most of the experiment setups especially in metaproteomics is often not available and cannot be easely applied. The unspecific assignment of enzymes to pathways in KEGG (also in other data repositories) that to not really belong to is not very surprising. This part is a bit overdosed and can be condensed dramatically. But while reading I came to the idea: Wouldn`t it be nice to consider not only pathway data but also other groups of lets say structural proteins, all proteins belonging to motility or to chloroplasts or other structures, enzyme classes - or lets say proteins belonging to groups which are defined in ontologies as such groups that can be summed up due to their membership in a group. Thsi would give you at least information of similar importance compared of summed up groups from pathway members. Implementation, mapping algorithm and availability information was sufficiently given and described except the missing availability for test purposes. My conclusion: I see the given tool as a nice gimmick in a preliminary draft state. I suggest condensing the text dramatically, remove redundancy, and to put some more efforts to make the tool available and a little more user friendly. After this submit the shortened manuscript as a note or short communication or whatever ist possible here Or Rewrite it, put much more efforts to improve the tools and make it a breathtaking application. and submit it as a research paper to show a direction where modern metaproteomics may go to.Author Response
Please see the attachment.

Round 2
Reviewer 2 Report
Nice work now, please try to improve visualization and include new pathway independent mapping (gene/protein assignment hierarchies, protein complexes, protein interaction networks) in future versions. For now I recommend the paper for publication.